# Fast Near-Field Frequency-Diverse Computational Imaging Based on End-to-End Deep-Learning Network

**DOI:** 10.3390/s22249771

**Published:** 2022-12-13

**Authors:** Zhenhua Wu, Fafa Zhao, Man Zhang, Sha Huan, Xueli Pan, Wei Chen, Lixia Yang

**Affiliations:** 1Information Materials and Intelligent Sensing Laboratory of Anhui Province, Anhui University, Hefei 230601, China; 2State Key Laboratory of Millimeter Waves, Southeast University, Nanjing 210096, China; 3East China Research Institute of Electronic Engineering, Hefei 230031, China; 4School of Electronics and Communication Engineering, Guangzhou University, Guangzhou 510006, China

**Keywords:** computational imaging, metasurface antennas, deep convolutional neural network, near field imaging

## Abstract

The ability to sculpt complex reference waves and probe diverse radiation field patterns have facilitated the rise of metasurface antennas, while there is still a compromise between the required wide operation band and the non-overlapping characteristic of radiation field patterns. Specifically, the current computational image formation process with a classic matched filter and other sparsity-driven algorithms would inevitably face the challenge of a relatively confined scene information sampling ratio and high computational complexity. In this paper, we marry the concepts of a deep convolutional neural network with computational imaging literature. Compared with the current matched filter and compressed sensing reconstruction technique, our proposal could handle a relatively high correlation of measurement modes and low scene sampling ratio. With the delicately trained reconstruction network, point-size objects and more complicated targets can both be quickly and accurately reconstructed. In addition, the unavoidable heavy computation burden and essential large operation frequency band can be effectively mitigated. The simulated experiments with measured radiation field data verify the effectiveness of the proposed method.

## 1. Introduction

With the capability to sculpt complex radiative wavefronts and couple energy from the reference wave to a desired radiation pattern, metasurface antennas have made tremendous progress in both antenna design and computational imaging (CI) applications [1,2,3]. In general, the ensemble imaging performance of a metasurface antenna computational imaging (MSACI) system is closely related to the optimized design of the frontend antenna in combination with the robust backend computational imaging algorithm. Since the first proof-of-concept study was demonstrated by Hunt et al. [4], numerous innovative metasurface architectures have been demonstrated, including metamaterial apertures, Mills–Cross cavity,  metallic leaky cavity, etc. The fundamental design of these metasurface antennas have evolved from the periodic uniform metamaterial aperture configuration to a chaotic slot arrangement. In the context of designing metasurface antennas for CI, key factors that influence CI performance, including the radiating elements, feeding structure and reconstruction algorithms. The intricate interplay of these factors in current a CI framework gives rise to some inherent inconsistencies between maximizing spatial frequency sampling and maintaining the necessary signal-to-noise: i.e., measurement quantity and quality, radiation losses, and the loaded Q-factor of the radiating element. Towards the primary goal of metasurface antenna CI, the trade-off between the abovedmentioned inconsistencies need to be properly balanced in practical imaging application.

Frequency-diverse imaging is a computational imaging technique that leverages frequency-diverse antennas to capture and reconstruct scene information. A frequency diversity antenna is actually a kind of super surface antenna; a frequency-diverse antenna radiates field patterns that vary strongly as a function of the driving frequency. Over a given frequency bandwidth, the fields of a frequency-diverse antenna vary spatially in a quasi-random (or quasi-orthogonal) manner across the operating frequency band. The key feature of frequency-diverse imaging is that the scene information is encoded onto these quasi-random field patterns, with the data acquired by means of a simple frequency sweep. Therefore, data acquisition is performed in an all-electronic manner (no mechanical scanning apparatus is required) without the need for phase shifting circuits.

In essence, the crux of MSACI lies in generating a sequence of spatially diverse field patterns to multiplex scene information and transfer the burden of image formation from hardware to software, yielding a plethora of work dedicated to the metasurface designing topic [5]. Moreover, in most of the current MSACI demonstrations including monostatic and multistatic configuration, the built measurement matrix (transfer function) is typically ill-conditioned and not generally subject to a straightforward matrix inversion. Scene reflectivities are then retrieved by using computational techniques mainly including least square solvers or compressive reconstruction techniques. These methods either directly adopt a matched filter (MF) approach or take the MF result as an initial estimate and make use of an iterative optimization method to refine the estimation [6]. In this way, the burden of imaging is thus shifted from antenna frontend to software backend, allowing constraints on the physical layer to be relaxed while maintaining high-fidelity and near real-time imaging. Specifically, the matching filtering(MF) algorithm in [1] is more frequently used in modern computational imaging systems when the imaging accuracy satisfies the requirements because of its low operation complexity and high efficiency. The Sparse Bayes learning(SBL) reconstruction algorithm in [7] is a sparse-based reconstruction algorithm that obtains imaging results closer to the real scene when the scene-echo signal-to-noise ratio is low. The primary challenge of these compressive sensing (CS) [8] reconstruction techniques is that the success of the reconstruction process depends heavily on the randomness as well as the condition number of the measurement matrix. Furthermore, when it comes to the imaging problem of extremely large dimensionality, the computation costs will drastically increase given the needed memory and storage usage for matrix inversion and iteration. Compressive reconstruction techniques with ill-conditioned measurement matrix still suffer from the prohibitively high signal-to-noise ratio to maintain a preferable imaging performance.

Note that deep learning (DL)-based strategies have now been successfully applied in a wide variety of imaging applications, including magnetic resonance imaging, X-ray computed tomography [9], computational optical imaging [10,11,12], and some cases of compressive meta-imagers [13,14]. Specifically, it has been found that they can typically outperform conventional image formation techniques in terms of image quality and computational speed [15,16,17,18,19,20,21]. More importantly, the intractable large dimensionality datasets and ill-conditioned features of a measurement matrix that are extremely troublesome to tackle for a CS-based approach can still be properly handled with a DL-based method [22]. For example, in order to address the challenging domain of inverse scattering imaging problems, a cascaded complex U-net model was presented in [23]. The CCU-net cascaded the Image Reconstruction Net (PRNet) and Phase Retrieval Net (IRNet) in order to reconstruct the target and recover the phase and amplitude of the dispersed field from the observed modulus of the total field, respectively. Numerical simulations showed that the suggested method could successfully recreate the scattering field from both simulated and real-world examples without any prior knowledge.

Provided prior knowledge about the nature of the scene to be imaged is available, these works have shown that it is possible to limit the number of sequential captures necessary for image reconstruction compared to the use of purely random patterns. Recent work in [24] took this idea even further by directly integrating a model of the physical layer into an artificial neural network in order to jointly learn optimal measurement and processing strategies based on a prior knowledge of scene, task, and measurement constraints. Since this “learned sensing” strategy enables one to minimize the acquisition of task-irrelevant information, it is highly task-specific and requires a supervised learning technique. In contrast to these prior works [23,25,26], the approach we propose here applies a deep network to the sensing matrix rather than the specific expected scene. Consequently, our approach is not scene dependent and does not require the use of sequential measurements relying on active reconfigurable antennas. We propose a system-dependent but scene-independent method relying on a frequency-sweep to generate a succession of random illumination patterns that interrogate the scene to be imaged; via the optimized deep neural reconstruction network, the dimensionality of the measurement matrix can be limited, and thereby the computational complexity of the image reconstruction can be reduced. Moreover, from the aspect of imaging system implementation, with the large needed operation frequency band and optimal designing burden of the current metasurface antenna frontend could be eased to some extent.

To lower the frequency-diverse CI reconstruction problem’s computational complexity and relieve the metasurface antenna frontend designing pressure, is at the heart of the present paper: we bring the concepts of convolutional neural networks (CNN) [27] into a microwave computational imaging [28,29] framework; we use a carefully designed light deep neural network to reduce the computational complexity of the imaging problem for real-time operation. For the delicate design of a fully connected layer and several convolutional layers, an adaptive momentum estimation (Adam) optimizer [30] with an adjusted learning rate, relatively low memory usage, and fast convergence speed is used to update the network weight parameters. The widely used sparse targets MNIST [27] datasets and complicated targets FMNIST [31] data are input to train the network. The trained reconstruction network is then able to directly retrieve scene reflectivities with high fidelity and handle the prohibitively large dimensionality of the measurement matrix. Extensive imaging simulations with measured radiation fields data are conducted to verify the effectiveness and robustness of our MSACI-Net method. In this paper, compared with a current matched filter and a compressed sensing reconstruction technique, our proposal could handle a relatively high correlation of measurement modes and a low scene sampling ratio. Since the antenna measurement modes are frequency-dependent, the superior image reconstruction capability could therefore reduce the need for a large operation bandwidth and more importantly, the antenna radiation efficiency could be enhanced in the antenna design process to some extent so as to maintain a relatively high SNR level, which is crucial for frequency-diverse computational imaging.

## 2. Forward Mathematical Model

The near-field computational imaging system based on metasurface antennas initially analyzes the entire working mechanism of the system before constructing a suitable mathematical model. The basic concept of metasurface antenna imaging is to construct a subwavelength resonant aperture structure to regulate the polarization characteristics of electric and magnetic dipoles and produce different polarization characteristics by designing subwavelength basic resonant units with different geometric structures. As the driving frequency changes, the radiation field excited from different subsets of resonators changes. Objects within the scene scatter the incident fields, producing the backscattered components detected by the waveguide probe at the transmitting antenna plane.

To a large extent, the concept of MSACI belongs to the inverse scattering problem. The frequency measurements collected by the receiving probe are related to the scene reflectivity through the measurement matrix (transfer function), which is the product of the electric fields from the transmitting antenna and the receiving probe at each location in the scene. Figure 1 shows the schematic diagram of a metasurface antenna imaging system with single transmitter and single receiver [32]. The total fields that propagate into the OEWG have the following proportionality:(1)$$g\left(f\right)\propto \int_{V}E_{\mathrm{TX}}\left({\vec{r}}^{\prime };f\right)\cdot E_{\mathrm{RX}}\left({\vec{r}}^{\prime };f\right)\cdot \sigma \left({\vec{r}}^{\prime }\right)\;d^{3}{\vec{r}}^{\prime }$$
where $E_{\mathrm{TX}}\left({\vec{r}}^{\prime }\right)$ and $E_{\mathrm{RX}}\left({\vec{r}}^{\prime }\right)$ denote the transmitted and received fields, respectively, and σ represents scene target reflectivities. Since the system is both diffraction and bandwidth limited, (Equation 1) can be written in a more general and concise matrix equation:(2)g=Hσ+n
where $g\in C^{M\times 1}$ is the receiving measurement vector collected by the low-gain OEWG, $\sigma \in C^{N\times 1}$ denotes the unknown reflectivity vector of the discretized scene space, $n\in C^{M\times 1}$ is an additive noise term included for generality, and $H\in C^{M\times N}$ is the measurement matrix constructed with transmitting and scattering fields.

In the vast majority of Fresnel-zone MSACI systems, to characterize the transmitting aperture, the near field of the aperture is raster scanned with a reference antenna, and the measured fields are then propagated to arbitrary locations in the scene using equivalent surface principles and the corresponding Green’s functions. To reconstruct the scene information, the matrix equation described in (Equation 2) needs to be solved for σ. In classic matrix theory, various matrix inversion techniques have been proposed for both well- and ill-conditioned H.

In the simplest implementation, assuming additive white Gaussian noise, an MF reconstruction suitably solves (Equation 2) as ${\hat{\sigma }}_{est}=H^{†}g$, where † denotes the conjugate transpose operator. Assuming the measurement value y of compressed sensing, the output is the original signal σ, and the measurement matrix is H, the original signal σ can be recovered from the measurements g by the neural network.
(3)$$\sigma =F_{w}\left(g\right)$$
(4)$$\theta =\mathrm{argmin}{∥\sigma -\hat{\sigma }∥}^{2}$$
where F(·) is the simplified neural network function, $\hat{\sigma }$ is the predicted value of the neural network output, θ is the resulting parameters to be optimized, w is the parameter weight, and the gradient descent method is used to minimize the root mean square loss function to train the network.Given the easy implementation capability, the matched filter approach is commonly applied in current MSACI using arbitrary field patterns. Note that when H is extremely underdetermined M≪N, the estimation with an MF method of under-sampled scenes could be further refined with more sophisticated reconstruction algorithms. In the following section, the CNN-based MSACI principle and network model are specifically introduced.

## 3. CNN-Based Computational Imaging

### 3.1. Imaging System Architecture

Figure 1 shows the structure of a near-field MSACI system architecture built using convolutional neural networks, named MSACI-Net. The linear mapping network in this section adopts a fully connected layer, which contains *N* neurons. The output of the $k_{x}\times k_{y}$ feature map is obtained when the measured values are used as network input, where $k_{x}$ and $k_{y}$ represent the horizontal and vertical near-field scan points. The collected measurements g and the MF method estimation results ${\hat{\sigma }}_{est}$ constitute the training dataset for the network system.

The fully connected layer is connected after the convolutional layer. In our demonstration, the size of the feature maps generated by all layers is consistent with the number of voxels to be imaged, the first and third layers use a 5×5 convolution kernel to reduce network parameters, and the number of channels is 30 and 60, respectively. The second and last layers of MSACI-Net use the 3×3 convolution kernel to generate the feature map as the output of the network, and the number of channels is 30 and 1, respectively. In this process, each layer of the convolutional layer performs a zero-padding operation to ensure that the size of the feature maps output in each layer remains the same.

### 3.2. Imaging Algorithm

The Adam algorithm is currently the most commonly used gradient optimization algorithm, which can handle sparse gradients and stationary objectives well. In particular, it can adaptively calculate learning rates for various parameters. Based on the ADAM algorithm, combined with our actual imaging problem, Algorithm 1 presents the optimized update process of our training data in MSACI-Net.
**Algorithm 1** MSACI-Net training optimization algorithm**Input:** H: measurement matrix; f(·): stochastic objective function with parameters; **N** training datasets $\sum_{i}^{N}\left\{g_{i},\sigma_{i}\right\}$; Number of iterations t=0; The exponential decay rate of the moment estimate $\rho_{1}=0.9$, $\rho_{2}=0.999$; Numerical stability constant $\varepsilon =10^{-8}$; Stepsize $\delta =10^{-3}$; initial parameter vector: $\theta_{0}$.1:**Iteration via a gradient descent scheme:**2:Random initialize weight parameters w and bias parameters b; Initialize first moment vector s←0; second moment vector r←0;3:**while θ does not converged**4:Compute gradient $g=\frac{1}{N}\nabla_{\theta }\sum_{i}^{N}L\left(f\left(\sigma_{i};\theta \right),g_{i}\right)$; t=t+1;5:Bias-corrected first moment and second moment estimate is revised $\hat{s}=\frac{s}{{1-\rho }_{1}^{t}}$; $\hat{r}=\frac{r}{{1-\rho }_{2}^{t}}$;6:Updated parameters $\theta =\theta_{0}+\frac{\delta \hat{s}}{\sqrt{\hat{r}}+\varepsilon }$,7:endwhile**Output:** Resulting parameters θ

The training process makes use of two learning methods of forwarding propagation and backpropagation to optimize the weights. The initial reconstruction result of the target in forward propagation is used as input and then output to each convolutional layer. In forward propagation, the output of the lth layer is: $g^{l}=f\left(u^{l}\right)$, where f(·) represents the activation function. All layers use Relu as the activation function, where the activation function can be expressed as:(5)$$Relu\left(u\right)=\left\{\begin{array}{c} u,u\geq 0\\ 0,u<0 \end{array}\right.$$

This function takes 0 when the input is less than 0 and takes the input value when the input is greater than or equal to 0.

Backpropagation is the process of correcting the weight coefficients and bias coefficients of the convolutional neural network through the output objective function. The specific correction process is as follows. The predict output of the convolutional neural network and the real target in the training set are, respectively, denoted as $\hat{\sigma }$ and σ. The cost function is:(6)$$E=\frac{1}{2}\sum {\left(\sigma -\hat{\sigma }\right)}^{2}$$

The weight coefficients in each layer are adjusted inversely, including weight coefficients and bias coefficients, according to the output error of each sample. This process calculates the partial derivative of the cost function that corresponds to each weight in the convolutional neural network. The sensitivity of the convolution layer is defined as $\delta =\frac{\delta E}{\delta u}$. The sensitivity of the non-input lth layer is:(7)$$\delta^{l}={\left(w^{l+1}\right)}^{T}\delta^{l+1}\cdot f{}^{\prime }\left(u^{l}\right)$$

The sensitivity of the output layer L is:(8)$$\delta^{L}=f^{\prime }\left(u^{L}\right)\cdot \left(\hat{\sigma }-\sigma \right)$$
where given $\frac{\partial u}{\partial b}=1$, $\frac{\partial E}{\partial b}=\frac{\partial E}{\partial u}\frac{\partial u}{\partial b}=\frac{\partial E}{\partial u}=\delta$. Therefore, the partial derivative of the bias coefficient is equal to the sensitivity of the corresponding convolutional layer.

Similarly, the partial derivative of the weight coefficient of the *l*-th layer is
(9)$$\frac{\partial E}{\partial w^{l}}=\sigma^{l-1}{\left(\delta^{l}\right)}^{T}$$

Finally, the weights are updated iteratively according to the Adam algorithm and Equations (5)–(7). The specific update methods are shown in (8) and (9).
(10)$$w_{new}^{l}=w^{l}-\eta_{1}\cdot \frac{\partial E}{\partial w^{l}}-\eta_{2}w^{l}$$
(11)$$b_{new}^{l}=b^{l}-\eta_{1}\cdot \frac{\partial E}{\partial b^{l}}-\eta_{2}b^{l}$$
where $\eta_{1}$ and $\eta_{2}$ are the gradient descent coefficient and the learning rate, respectively.

In addition, MSE and PSNR are selected to evaluate accuracy and image quality. The solution process of MSE is shown in the following equation:(12)$$\mathrm{MSE}=\frac{\sum_{i=1}^{m}\sum_{j=1}^{m}(\sigma \left(i,j\right)-\hat{\sigma }{\left(i,j\right))}^{2}}{m}$$
where *m* is the number of images, and $\sigma_{i}$ and $hat\sigma_{i}$ severally represent the pixel values of the original image and the reconstructed image at (i,j). PSNR is essentially the same as MSE. The higher the PSNR value, the smaller the difference between the two contrast images. In the subsequent dataset construction, we will normalize the scene target image; all target images have scattering coefficients between 0∼1. Therefore, the PSNR formula is given as:(13)$$\mathrm{PSNR}=10\log\frac{1\times 1}{\mathrm{MSE}}$$

Furthermore, Algorithm 2 illustrates the specific content of the near-field computational imaging algorithm of the metasurface antenna based on the convolutional neural network, which principally incorporates the following steps. First, according to the given metasurface antenna system parameters, combined with the classical image datasets MNIST and Fashion-MNIST, a certain number of training sets, test sets, and validation sets are generated, respectively. Then, set the basic parameters of the convolutional neural network in the algorithm, such as the number of fully connected layers are three, the number of convolutional layers are four, the size of the convolution kernel of each layer, and the number of channels of the convolutional network in each layer. Select the appropriate gradient optimization training algorithm, this paper selects Algorithm 1, and then sets the learning rate, gradient descent coefficient, training period, and the number of samples participating in the training of the gradient optimization algorithm. After completing the above preparations, we utilize Algorithm 1 to train the network. The weight coefficients of the convolutional layer are continuously updated during the forward propagation and back propagation until the end of the training period. At this time, the loss function of the system tends to converge, and the trained network model is conserved to facilitate subsequent testing of the measured data.
**Algorithm 2** MSACI-Net reconstruction algorithm**Input:** The train dataset $\left(g_{\mathrm{tr}},\sigma_{\mathrm{tr}}\right)$, test dataset $\left(g_{t}\right)$, weight parameter: θ, maximum iteration number: **T**;1:Randomly initialize the weight parameter W of all layers of the MSACI-Net network;2:**for** T **do**3:      Batch $g_{\mathrm{tr}}$ and $\sigma_{\mathrm{tr}}$ of the training set are input to MSACI-Net as the input and output of the network respectively.4:      Utilization Equation (6) as loss function, and algorithm 1 above is used to train the weight of the network.5:      After T cycles, preserve the trained network model.6:      Input the echo data $g_{t}$ of the test set, call the above trained network model for testing, and retrieve the reconstruction target $\sigma_{t}$.7:**end for****Output:** The reconstructed result $\sigma_{t}$.

## 4. Imaging Simulations with Measured Fields Data

### 4.1. Measured Fields Data

In this section, we verify the validity of the proposed MSACI-Net method with measured radiation fields data. For near-field CI capability demonstration purpose, in this subsection, a two-dimensional parallel plate waveguide metasurface antenna with waveguide slot feeding mechanism is designed and fabricated. The frequency-diverse radiation fields (measurement matrix) are measured through near-field scanning, and the scanning plane is 0.5 m away from antenna platform. The measured measurement matrix is then used to perform image reconstruction experiments. To ensure that an ample backscattered signal is collected from all possible directions and at all frequencies, an open-ended waveguide (OEWG) probe is used as receiving antenna, comprising a panel-to-probe configuration. The panel size of the antenna is 250×250 mm^2^, the dielectric constant is 3.66, and the loss tangent value is 0.003. The thickness of the substrate between the copper ground plane and the conductive copper metamaterial hole is 0.5 mm, the upper conductor of the waveguide adopts 125 × 125 cELC metamaterial resonators, and the Q-factor of each resonator is between 50∼60.

The system parameters of the antenna are shown in Table 1, and the imaging experiment is based on the simulated metasurface antenna radiation field pattern data and the imaging scene. The operation bandwidth of the antenna is 33∼37 GHz, the frequency sampling interval is 10 MHz, and the pattern pattern of each frequency point is sampled along the two-dimensional spherical coordinate system of elevation and azimuth. The field of view (FOV) size is elevation (−60∼60°), and the sampling interval is 2${}^{{}^{\circ}}$; the azimuth sampling line of sight is (−60∼60°), and the sampling interval is 2${}^{\circ }$, so the dimension of the original pattern T is 400 × (61 × 61). In the scheme of feeding by waveguide slot coupling, the return loss and radiation efficiency are shown in Figure 2 and Figure 3.

The original target that contains the sparse target and extended target are employed to qualitatively evaluate the imaging ability of the measurement matrix for the scene. In order to qualitatively evaluate the ability of the measurement matrix to image the scene, the image containing the point scattering target in the same dimension as the measurement matrix is used as the original image. It should be emphasized that the measurement matrix at this time is the pattern data, not the metamaterial in the actual imaging space. The measurement matrix is formed by the radiated field of the aperture antenna. Figure 4a–d show the metasurface antenna feeding mechanism and radiation fields scanning plane.

### 4.2. Data Preparation

The received echo vector in the imaging process can be regarded as mapping of the scene reflectivity function. This mapping is nonlinear, given the complex scattering of electromagnetic waves. We aim to use CNN in this deep learning imaging framework to iteratively learn the nonlinear mapping of this physical model. The training samples of the network are directly obtained from the original antenna pattern. Specifically, the original measurement value may be used as the input, and the output is the target image of the scene. Compared with the traditional method, the use of the deep neural network method to reconstruct the target image can provide a more realistic and accurate physical model representation.

In the simulation experiment, we choose the classic datasets MNIST and FMNIST to build our target dataset. Both datasets contain 70,000 images, of which 60,000 images form the training set and 10,000 images form the test set. According to the needs of the experiment, we adjust the original image size in the dataset from 28 × 28 to 61 × 61 and regard it as an image that is composed of multiple scattering points with random values between (0, 1) scattering coefficients as the output network of the original target scene. The input value is the echo measurement value of the target scene that passes through the metasurface antenna pattern. The samples used in the experiment are formed through this process. Each group of samples contains a measurement value sample and an original image. The datasets are severally named MSACI-MNIST and MSACI-FMNIST, and they are divided according to the training set that accounts for 70%, the validation set that occupies 20% and the test set that accounts for 10%. The mean square error (MSE) is used as the loss function, and the Adam optimization algorithm is used for optimization. The learning rate is 0.001, and the batch size is 128. The model is trained for 100 cycles. After each training cycle, the validation data is randomly selected to generate a verification set to monitor network performance. This stage is very time-consuming. When the network training is completed, we save the network model and input the scene echo signal into the network to obtain the imaging result of the target in real time. The final prediction is kept on the test set.

The model is implemented in the Python programming environment using the Keras deep learning framework and TensorFlow backend platform. The model is then trained on a computer with GPU NVIDIA 3070Ti and CUDA version 11.0. The batch processing of the dataset is completed on the MATLAB platform. In addition, all reconstruction results are obtained on a computer equipped with Intel(R) Core(TM)i7-10700 CPU. Without using the GPU, the training process for the 100 epochs of our imaging network model using the constructed dataset takes 1.5 h. The same number of training epochs, however, takes around 20 min after calling the GPU.

### 4.3. Numerical Tests

Our MSACI-Net algorithm based on the CNN framework is trained on the ordinary sparse dataset MSACI-MNIST and the extended target dataset MSACI-FMNIST. After completing the training, we conduct numerical experiments and performance evaluation to evaluate the performance of our algorithm. To assess the effectiveness of the algorithm, we implement an experiment with different scene targets when the number of measurement modes is 400. The scene images come from the test sets of MSACI-MNIST and MSACI-FMNIST datasets. We select 12 target images from the two 2 sets to conduct the experiments. The original sparsity of these 12 targets are 0.6312, 0.5954, 0.6721, 0.6856, 0.6423, 0.6229, 0.5117, 0.4996, 0.6463, 0.4786, 0.5116, and 0.4523, respectively. It should be emphasized that noise-free scenes are considered throughout the imaging simulation process to focus on the implementation of the proposed algorithm, and the test results are shown in Figure 5. The experimental results show that our MSACI-Net can not only reconstruct ordinary sparse targets, but also reconstruct extended targets under the conditions of a fixed number of measurement modes and scene compression ratio. Moreover, reconstructed images also can be obtained with high quality by the proposed algorithm.

In the MSACI-Net of our proposal, the ADAM algorithm is chosen to train the gradients, where the learning rate is chosen to be $1\times 10^{-3}$, which can better control the weight update rate and allow the training to converge to better performance. The number of learning cycles is 100 epochs, considering that the increase in the number of training times can make the network training more mature and finally show better imaging performance.

To further evaluate the performance of the proposed algorithm, we set the experimental conditions under different compression ratios of 0.1, 0.05, and 0.025 separately. We also use the traditional MF method, an iterative sparse Bayesian learning (SBL) algorithm and the U-Net method [23] for the two sets of targets and conduct comparative experiments with the proposed algorithm under the same conditions. The experimental results of the two groups are shown in Figure 5 and Figure 6.

The experimental results show that our MSACI-Net can reconstruct the target image clearly when the scene information sampling ratio is 0.1. Our algorithm can effectively reconstruct the basic shape of the scene target with the decrease in the scene information sampling ratio, even the sampling ratio of 0.025. In contrast, the MF and the SBL algorithm can no longer work normally when the scene information sampling ratio is 0.1. The reconstructed target image is not acceptable, and the original shape cannot be distinguished. In Figure 6 and Figure 7, although the U-Net reconstruction imaging results showed the approximate shape, they could not capture the detailed features specific to the target. Moreover, the original target contour features cannot be precisely reconstructed in the SBL algorithm imaging results; the SBL algorithm is unable to operate in the case of extremely low compression ratios, and the imaging outcomes essentially show no details of the original target. Overall, the performance of the proposed algorithm is more robust and effective than traditional algorithms in low scene compression ratio and high coherence measurement mode. The quality of the reconstructed target image is also high and can face the challenges of imaging algorithm performance in complex scene conditions.

We employ MSE and PSNR to quantitatively evaluate the accuracy of imaging reconstruction and the quality of the imaging results. Quantitatively, the MSE in MSACI-MNIST and MSACI-FMNIST of different methods under different information compression ratios are calculated, and the results are shown in Figure 8a and Figure 9a.The PSNRs in MSACI-MNIST and MSACI-FMNIST of different methods under different information compression ratios are also calculated, and the results are shown in Figure 8b and Figure 9b. As can be seen, the proposed algorithm gradually decreases in MSE when the scene information sampling ratio increases. Our proposed algorithm is numerically lower than the MF and the SBL algorithm in this standard. Compared with the generally applied CS method, our MSACI-Net algorithm shows preferable imaging capability given that the end-to-end neural network can adaptively adjust imaging error. The quality of the image reconstruction with our algorithm has significant improvement in PSNR with the same scene information sampling ratio.

We also record the time spent by the two algorithms in the reconstruction of the scene target. The method based on a convolutional neural network is easy to parallelize during imaging reconstruction. Thus, we record the reconstruction time of the MSACI-Net algorithm with CPU and GPU. Table 2 shows the average running time of the 100 epochs test process with the two algorithms on different targets. It is obvious that the proposed algorithm takes less time than the MF and SBL algorithm. Therefore, this method is superior in terms of imaging efficiency and can even achieve near real-time imaging reconstruction.

## 5. Discussion

Compared with the generally applied MF method, our MSACI-Net algorithm shows preferable imaging capability under no-noise conditions, given that the end-to-end neural network can adaptively adjust imaging error. In order to further test the anti-noise and robustness of the proposed algorithm, we add additive noise when generating the dataset. Thus, the scene information sampling ratio in the MSACI-MNIST and MSACI-FMNIST datasets are set to 0.1. To study the impact of noise and noise on network performance, we divide our dataset into three noise-free cases, namely, SNR = 0 dB, SNR = 5 dB, and SNR = 10 dB after 100 rounds of testing. Qualitative and quantitative tests are conducted on these two datasets. The imaging reconstruction results are shown in Figure 10. The results show that our MSACI-Net algorithm can not only reconstruct ordinary sparse targets, but also reconstruct extended targets under the conditions of a different SNR and fixed scene compression ratio. Moreover, the proposed algorithm can also obtain reconstructed images with high quality and has relatively excellent anti-noise performance and good robustness when the scene information sampling ratio is 0.1.

Quantitatively, the MSEs of different methods under different SNRs are calculated, and the results are drawn in Figure 11a,b. These results show that our network can withstand the test with or without noise in the training data, with negligible impact on accuracy when reconstructing the target. Overall, our proposed algorithm compensates for its shortcomings when performing near-field computational imaging under the condition of insufficient frontend hardware design of the antenna. This result shows excellent effectiveness and robustness, which indicates the promising potential of the MSACI tool.

## 6. Conclusions

In this paper, the near-field metamaterial aperture CI with CNN-based method is demonstrated. Compared to current MF and CS reconstruction techniques, our delicately trained deep network could handle both sparse and complicated scene targets scenarios under a relatively low scene information sampling ratio and SNR levels, yielding a relatively narrow needed operation frequency band and alleviating the optimal designing burden of a metasurface antenna frontend. The trained network weight parameters can be readily accessed to deal with radiation fields data obtained with different types of current metasurface antennas. Moreover, the proposed CNN-based reconstruction approach could facilitate far-range imaging since the randomness feature among radiation field patterns could be sacrificed to a certain extent to meet the requirement of antenna radiation efficiency. In future work, we intend to improve the network performance, speed up the imaging rate, improve the imaging quality, and with training datasets and echo signal experiment to verify our method. In addition, we have begun to prepare the proposed method for far-field computing imaging and near-field three-dimensional imaging research and through the measured data to verify the imaging performance of our network.

## Figures and Tables

**Figure 1 sensors-22-09771-f001:**
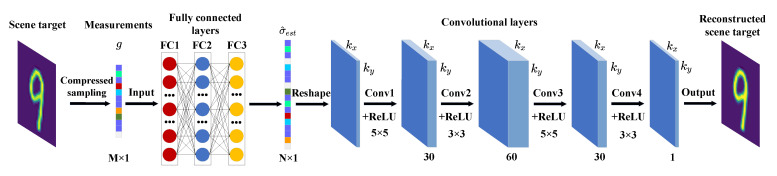
MSACI-Net system structure diagram.

**Figure 2 sensors-22-09771-f002:**
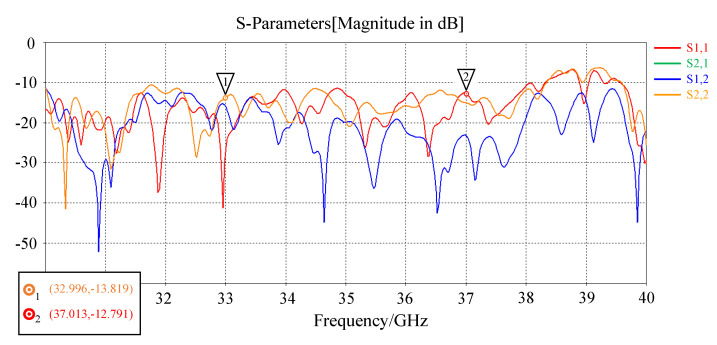
Antenna S-parameters [magnitude in dB].

**Figure 3 sensors-22-09771-f003:**
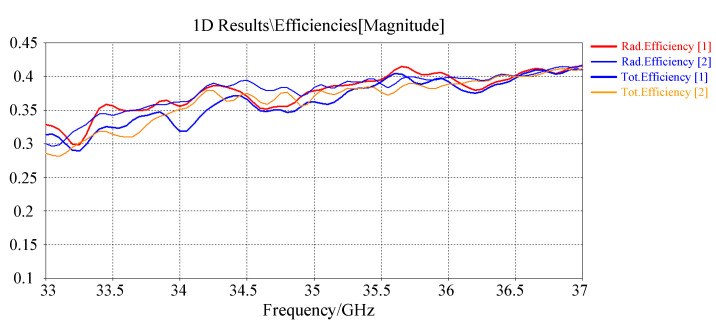
Antenna radiation efficiency [magnitude].

**Figure 4 sensors-22-09771-f004:**
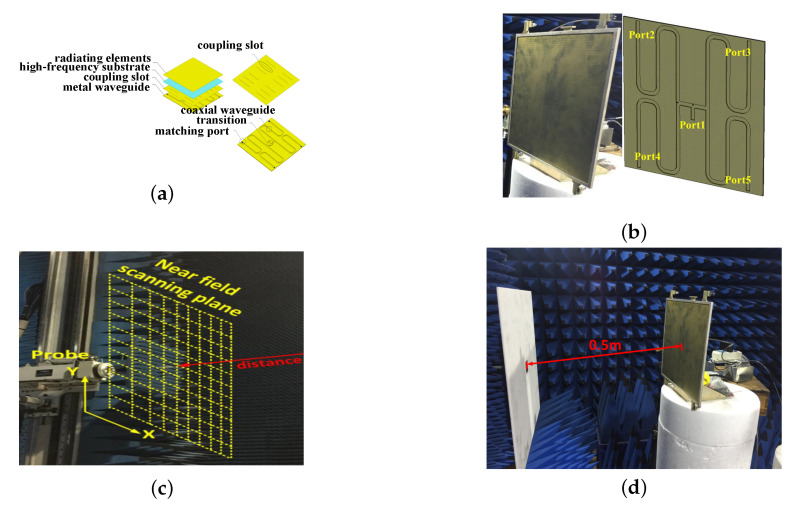
(**a**) Waveguide slot feeding mechanism; (**b**) antenna prototype; (**c**) radiation fields scanning plane.; (**d**) measurement matrix characterization.

**Figure 5 sensors-22-09771-f005:**
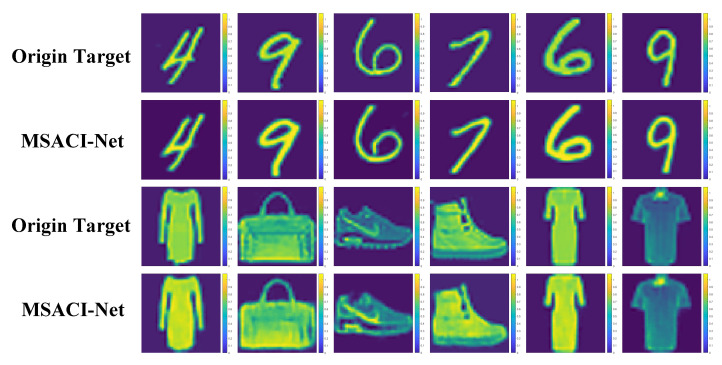
Reconstruction results from MSACI-Net with different scene target.

**Figure 6 sensors-22-09771-f006:**
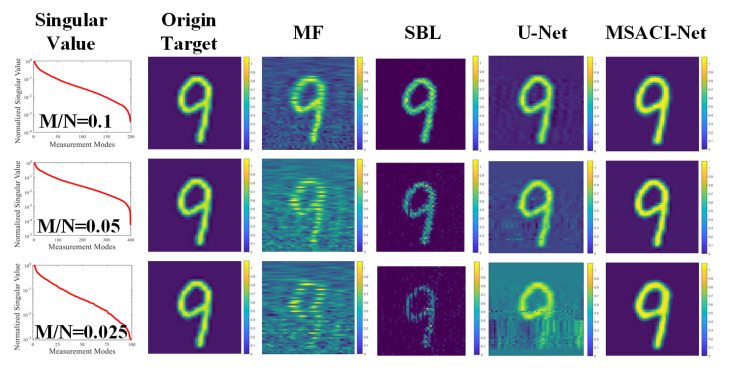
MSACI-MNIST: Reconstruction results in four imaging algorithms with different scene information sampling ratios.

**Figure 7 sensors-22-09771-f007:**
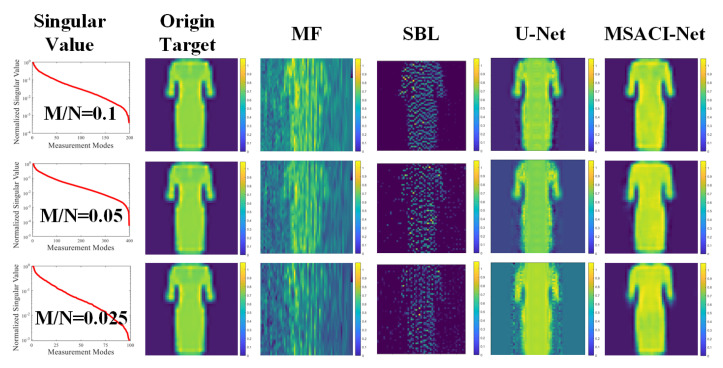
MSACI-FMNIST: Reconstruction results in four imaging algorithm with different scene information sampling ratios.

**Figure 8 sensors-22-09771-f008:**
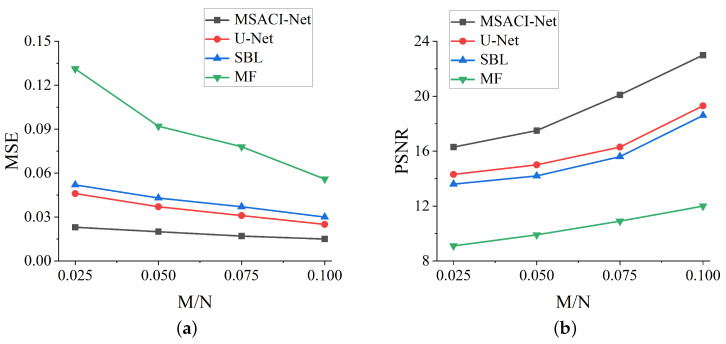
MSACI-MNIST: Imaging results with different scene information sampling ratio: (**a**) MSE performance; (**b**) PSNR performance.

**Figure 9 sensors-22-09771-f009:**
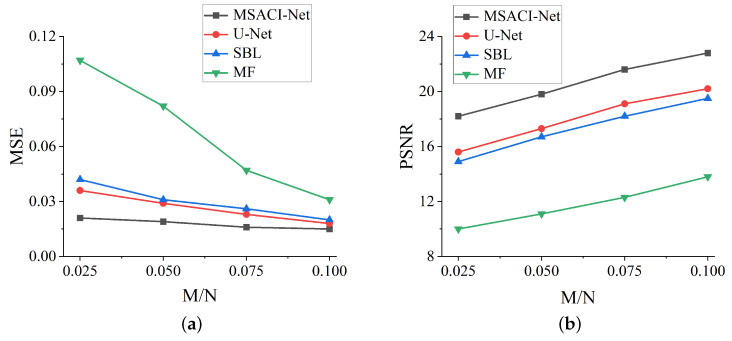
MSACI-FMNIST: Imaging results with different scene information sampling ratio: (**a**) MSE performance; (**b**) PSNR performance.

**Figure 10 sensors-22-09771-f010:**
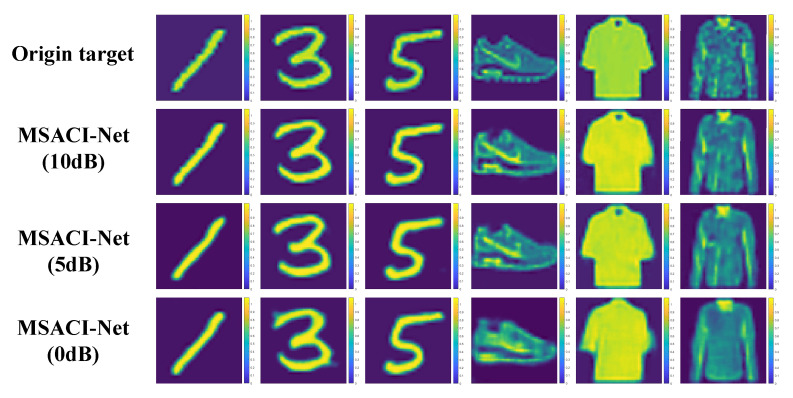
Reconstruction results from MSACI-Net with different SNR.

**Figure 11 sensors-22-09771-f011:**
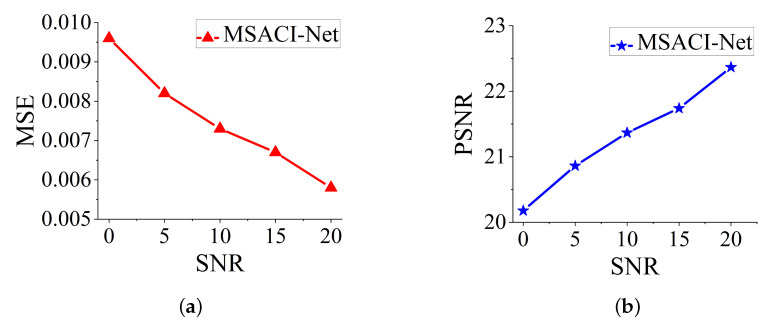
MSACI-FMNIST: Imaging results with different scene information sampling ratio: (**a**) MSE performance; (**b**) PSNR performance.

**Table 1 sensors-22-09771-t001:** Main system parameters of metasurface antenna.

Parameters	Values
Operation bandwidth	33∼37 GHz
Antenna panel size	250×250 mm^2^
Number of resonance units	125×125
Frequency sampling interval	10 MHz
Field of view (Azimuth)	−60∼60°
Field of view (Elevation)	−60∼60°
Azimuth sampling interval	$2^{\circ }$
Elevation sampling interval	$2^{\circ }$
Dimensions of **T**	400×3721

**Table 2 sensors-22-09771-t002:** Imaging runtime.

Methods	Values
M F	2.1756 s
SBL	3.5184 s
U-Net	0.4014 s
MSACI-Net(CPU)	0.2785 s
MSACI-Net(GPU)	0.0584 s

## Data Availability

The data presented in this study are available on request from the corresponding author.

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
