# Peer review of "Fast Near-Field Frequency-Diverse Computational Imaging Based on End-to-End Deep-Learning Network"

_sensors, 2022, doi:10.3390/s22249771_

Round 1

Reviewer 1 Report (Previous Reviewer 1)

This paper is fully revised and looks much better than the original version. I recommend this paper be accepted. 

Author Response

Reviewer 2 Report (New Reviewer)

The paper presents an interesting proposal, which is shown to exhibit better performance to some previously proposed schemes.

The authors are suggested to give more detailed review of prior solutions (state-of-the-art review), and explain why the exact reference methods are used in comparison analysis with their proposed method (scheme). The current presentation just give superficial overview in the introduction section.

Author Response

Reviewer 3 Report (Previous Reviewer 2)

This manuscript is a revised version that was rejected before. I can see that the authors have fully considered the reviewers' comments and suggestions, and made significant modifications. I think all the technical issues have been well addressed. Hence, I suggest this revised version can be accepted for publication in Sensors.

Author Response

This manuscript is a resubmission of an earlier submission. The following is a list of the peer review reports and author responses from that submission.

Round 1

Reviewer 1 Report

This paper is not well organized. It is very confused reading this paper:

1.      Some concepts are not fully explained. For example, what is frequency-diverse computational imaging? What is the relationship between this imaging and metasurface antennas? Why could the proposed method solve the facing challenge?

2.      Figure 1 is confusing since the Scene target and the Reconstructed scene target are the same. Figure 2 is very unclear.

3.      What are MNIST and FMNIST. What is the meaning of Fig. 3, do the authors use the database from the literature or captured by themselves?

4.      There is a lack of analysis and discussion on the kernel parameters of the network.

5.      In Figs. 4-6, the input figures should be shown as well.

Besides, in this experiment part, I cannot see clear connections between the results and the main purpose of the paper. 

Reviewer 2 Report

This manuscript proposed a novel end-to-end deep learning network for fast near-field frequency diverse computational imaging, based on which point-size objects and more complicated targets can both be fastly and accurately reconstructed. To validate the performance of the proposed netowrk, the simulated experiments was conducted, with satisfactory results. Overall, the topic of this research is interesting, and the manuscript was well organised and written. The detailed comments are summarised as follows.

1.       The contribution and innovation of the manuscript should be clarified clearly in abstract and introduction.

2.       Please broaden and update the literature review on real applications of CNN or deep learning methods. E.g. Crack detection of concrete structures using deep convolutional neural networks optimized by enhanced chicken swarm algorithm. A novel deep learning-based method for damage identification of smart building structures.

3.       How did the authors set the hyperparameters of the proposed network to achieve the optimal model performance?

4.       Training time of the proposed method should be provided as well.

5.       The proposed algorithm has not been presented convincingly about its advantages.

6.       More future research should be included in conclusion part.
